

# Fingerprints of the COVID-19 economic downturn and recovery on ozone anomalies at high-elevation sites in North America and Western Europe

Davide Putero[1], Paolo Cristofanelli[2], Kai-Lan Chang[3], Gaëlle Dufour[4], Gregory Beachley[5], Cédric Couret[6], Peter Effertz[7], Daniel A. Jaffe[8], Dagmar Kubistin[9], Jason Lynch[5], Irina Petropavlovskikh[7], Melissa Puchalski[5], Timothy Sharac[5], Barkley C. Sive[10], Martin Steinbacher[11], Carlos Torres[12], and Owen R. Cooper[3]

[1]National Research Council of Italy – Institute of Atmospheric Sciences and Climate, CNR–ISAC, Turin, Italy
[2]National Research Council of Italy – Institute of Atmospheric Sciences and Climate, CNR–ISAC, Bologna, Italy
[3]Cooperative Institute for Research in Environmental Sciences, University of Colorado Boulder/NOAA Chemical Sciences Laboratory, Boulder, U.S.A.
[4]Université de Paris Cité and Univ. Paris Est Créteil, CNRS, LISA, Paris, France
[5]Office of Atmospheric Protection, U.S. Environmental Protection Agency, Washington DC, U.S.A.
[6]German Environment Agency, Zugspitze, Germany
[7]Cooperative Institute for Research in Environmental Sciences, University of Colorado Boulder/NOAA Global Monitoring Laboratory, Boulder, U.S.A.
[8]University of Washington, School of STEM/Department of Atmospheric Sciences, Bothell/Seattle, U.S.A.
[9]Hohenpeißenberg Meteorological Observatory, Deutscher Wetterdienst, Hohenpeißenberg, Germany
[10]Air Resources Division, National Park Service, Denver, U.S.A.
[11]Empa, Laboratory for Air Pollution & Environmental Technology, Dübendorf, Switzerland
[12]Izaña Atmospheric Research Center, State Meteorological Agency of Spain, IARC-AEMET, Tenerife, Spain

**Correspondence:** Davide Putero (d.putero@isac.cnr.it)

**Abstract.** With a few exceptions, most studies on tropospheric ozone ($O_3$) variability during and following the COVID-19 economic downturn focused on high-emission regions or urban environments. In this work, we investigated the impact of the societal restriction measures during the COVID-19 pandemic on surface $O_3$ at several high-elevation sites across North America and Western Europe. Monthly $O_3$ anomalies were calculated for 2020 and 2021, with respect to the baseline period 2000–2019, to explore the impact of the economic downturn initiated in 2020 and its recovery in 2021. In total, 41 high-elevation sites were analyzed: 5 rural or mountaintop stations in Western Europe, 19 rural sites in the Western US, 4 sites in the Western US downwind of highly polluted source regions, 4 rural sites in the eastern US, plus 9 mountaintop or high-elevation sites outside Europe and the United States to provide a "global" reference. In 2020, the European high-elevation sites showed persistent negative surface $O_3$ anomalies during spring (March–May, i.e., MAM) and summer (June–August, i.e., JJA), except for April. The pattern was similar in 2021, except for June. The rural sites in the Western US showed similar behavior, with negative anomalies in MAM and JJA 2020 (except for August), and MAM 2021. The JJA 2021 seasonal average was influenced by strong positive anomalies in July, due to large and widespread wildfires across the Western US. The polluted sites in the Western US showed negative $O_3$ anomalies during MAM 2020, and a slight recovery in 2021, resulting in a positive average anomaly for MAM 2021 and a pronounced month-to-month variability in JJA 2021 anomalies. The Eastern US sites



were also characterized by below average $O_3$ for both MAM and JJA 2020, while in 2021 the negative values exhibited an opposite structure compared to the Western US sites, which were influenced by wildfires. Concerning the rest of the World, a global picture could not be drawn, as the sites, spanning a range of different environments, did not show consistent anomalies, with a few sites not experiencing any notable variation. Moreover, we also compared our surface anomalies to the variability of mid-tropospheric $O_3$ detected by the IASI satellite instrument. Negative anomalies were observed by IASI, consistent with
published satellite and modeling studies, suggesting that the anomalies can be largely attributed to the reduction of $O_3$ precursor emissions in 2020.

## 1   Introduction

Tropospheric ozone (hereinafter simply referred to as $O_3$) is a short-lived climate forcer (Szopa et al., 2021) that plays a key role in the climate system. It is one of the most powerful anthropogenic greenhouse gases (the third most important, after
carbon dioxide and methane), and it also impacts the lifetime of methane, which is one of the $O_3$ precursors (Monks et al., 2015; Gulev et al., 2021). Moreover, at the surface it also has adverse effects on ecosystems, crop productivity, and human health (Fleming et al., 2018; Mills et al., 2018).

The COrona VIrus Disease (COVID-19) pandemic emerged in late 2019, and initiated a global economic downturn in 2020, which was characterized by a drastic reduction of emissions related to several sectors, such as private transportation or domestic
and international aviation (e.g., Le Quéré et al., 2020; Friedlingstein et al., 2022). The reduction of emissions turned into a reduction of air pollutants that can directly be related to $O_3$ variability due to its photochemical formation from $O_3$ precursors, such as nitrogen oxides (NO and $NO_2$), carbon monoxide (CO) and non-methane volatile organic carbons (NM-VOCs).

Several studies in the past few years have investigated the impact of the COVID-19 economic downturn on $O_3$ concentrations and variability, at global, regional, and local scales (Gkatzelis et al., 2021; Sokhi et al., 2021). However, most of these works
focused on high-emission sources or urban environments (Sicard et al., 2020; Adam et al., 2021; Chossière et al., 2021; Keller et al., 2021). A number of studies indicated varying $O_3$ behavior as a function of the reduction in the emissions, mainly dependent on whether the photochemical $O_3$ formation in the considered regions was NOx- or VOC-limited (Gaubert et al., 2021; Matthias et al., 2021; Mertens et al., 2021; Cuesta et al., 2022).

Concerning free tropospheric values, which could be considered representative of background atmospheric conditions, Stein-
brecht et al. (2021) reported a reduction of $O_3$ of 7% (~4 ppb) from April to August 2020, in the 1–8 km altitude region of northern mid-latitudes, with respect to the 2000–2020 climatological mean. Cristofanelli et al. (2021) observed reductions in the 2020 monthly mean $O_3$ values (with respect to a 25-year climatological average) at a mountaintop site in Italy. Other studies have indicated the presence of negative $O_3$ anomalies in the free troposphere in 2020, mainly as a consequence of emissions reductions (Bouarar et al., 2021; Clark et al., 2021; Miyazaki et al., 2021). Chang et al. (2022) determined that the free tro-
pospheric $O_3$ negative anomalies in 2020 were the most profound since 1994 for both Europe and Western North America, and that the 2020 anomalies had a weakening influence on the 1994–2019 positive $O_3$ trends above these regions. The $O_3$ reductions in the free troposphere were also confirmed by the work of Ziemke et al. (2022), in which satellite measurements of





tropospheric column $O_3$ show that the 2020 negative anomalies in the Northern Hemisphere occurred again in spring–summer 2021.

In this study, we analyzed the $O_3$ variability at 41 high-elevation sites across the Globe, representative of different environments and emission source regions, during the COVID-19 economic downturn. The aim of this work is to determine if the negative $O_3$ anomalies observed in the free troposphere (e.g., Bouarar et al., 2021; Clark et al., 2021; Steinbrecht et al., 2021; Chang et al., 2022; Ziemke et al., 2022) also occurred in the boundary layer, by focusing on a selection of mountaintop and high-elevation monitoring sites, with available data up to December 2021. Therefore, our study will cover both the COVID-19

economic downturn in 2020, and the following year, 2021, which was associated with a recovery of emissions representative of a pre-pandemic level.

The paper is structured as follows. Section 2 will present the methodologies adopted, Sect. 3 will focus on the discussion of the results obtained, and conclusions will be drawn in Sect. 4.

## 2   Methods

### 2.1   Surface ozone

Figure 1 shows the geographical location of the sites considered in this study, and additional details (station name, latitude, longitude, and elevation) for each station are reported in Table 1. Hereinafter we will refer to each site by using its acronym (code), also listed in Table 1. The stations comprise a selection of 41 high-elevation sites worldwide, representative of five different environments (the so-called "regions" in Table 1 and Fig. 1), i.e.: (i) 5 European rural or mountaintop sites (EUR),

(ii) 19 sites in the Western US representative of rural conditions (WUS_R), (iii) 4 sites in the Western US downwind of highly polluted source areas (WUS_P), (iv) 4 eastern US rural sites (EUS), and (v) 9 other globally distributed mountaintop or high-elevation sites (OT). The OT sites are representative of very different environments (e.g., Antarctic conditions compared to the tropical latitudes of Mt. Kenya or Mauna Loa); however, these sites provide a characterization of baseline $O_3$ variability in several regions of the World that are far away from major anthropogenic emissions.

At all of the considered sites the UV-absorption method is used for measuring surface $O_3$, and common guidelines are followed for the reliability and consistency of $O_3$ data among the different monitoring programs (e.g., Galbally et al., 2013). With the exception of MBO, the 27 high-elevation monitoring sites in the US are Clean Air Status and Trends Network (CASTNET) sites maintained by the Environmental Protection Agency (EPA) and the National Park Service (NPS). The remaining 14 sites are all part of the Global Atmosphere Watch programme of the World Meteorological Organization (WMO/GAW), including

9 global stations, 4 regional stations and 1 contributing station. The data processing involved, when necessary, re-formatting the data, time shift to UTC (all measurements hereby presented refer to UTC), and unit conversions. Similar to the methods of Cooper et al. (2020) and Cristofanelli et al. (2020), the ZSF time series shown here is derived from merging the observations carried out both at Zugspitze summit and at the Schneefernerhaus station (see more details in the Supplementary Material).



**Table 1.** List of the stations used in this study for calculating the monthly anomalies, also reported in Fig. 1. Trend values ($50^{th}$ percentile, in ppb per decade) are calculated by using quantile regression, and reported together with 95% confidence intervals and $p$-values, computed by adopting the moving block bootstrap algorithm (see Sect. 2.1.2). Period indicates the range of years considered for the trend calculation. The region abbreviations are as follows: WUS_R = Western US "rural", WUS_P = Western US "polluted", EUS = Eastern US, EUR = Europe, OT = Other.

| Site name | Code | Lat. (° N) | Lon. (° E) | Elevation (m a.s.l.) | Region | Period | Trend (ppb per decade) |
|---|---|---|---|---|---|---|---|
| Canyonlands National Park | CAN | 38.46 | −109.82 | 1794 | WUS_R | 2000–2021 | $-1.87\,[\pm0.98], p < 0.01$ |
| Centennial | CNR | 41.36 | −106.24 | 3178 | WUS_R | 2000–2021 | $-1.65\,[\pm1.05], p < 0.01$ |
| Chiricahua National Monument | CNM | 32.01 | −109.39 | 1570 | WUS_R | 2000–2021 | $-1.13\,[\pm1.13], p = 0.05$ |
| Concordia | DCC | −75.10 | 123.33 | 3233 | OT | 2008–2021 | $0.32\,[\pm1.05], p = 0.55$ |
| Cranberry | PNF | 36.11 | −82.05 | 1219 | EUS | 2000–2021 | $-3.16\,[\pm1.37], p < 0.01$ |
| Craters of the Moon National Monument | CRA | 43.47 | −113.56 | 1815 | WUS_R | 2000–2021 | $-1.21\,[\pm1.34], p = 0.03$ |
| Denali National Park | DEN | 63.72 | −148.97 | 663 | OT | 2000–2021 | $0.19\,[\pm0.82], p = 0.65$ |
| Dinosaur National Monument | DIN | 40.44 | −109.30 | 1463 | WUS_R | 2007–2021 | $-0.27\,[\pm2.20], p = 0.81$ |
| El Tololo | TLL | −30.17 | −70.80 | 2154 | OT | 2000–2021 | $2.26\,[\pm0.58], p < 0.01$ |
| Glacier National Park | GNP | 48.51 | −114.00 | 976 | WUS_R | 2000–2021 | $1.30\,[\pm1.10], p = 0.02$ |
| Gothic | GTH | 38.96 | −106.99 | 2926 | WUS_R | 2000–2021 | $-1.56\,[\pm0.91], p < 0.01$ |
| Grand Canyon National Park | GRC | 36.06 | −112.18 | 2073 | WUS_R | 2000–2021 | $-2.06\,[\pm0.90], p < 0.01$ |
| Grand Teton National Park | GTP | 43.67 | −110.60 | 2105 | WUS_R | 2011–2021 | $-1.28\,[\pm2.74], p = 0.35$ |
| Great Basin National Park | GBN | 39.00 | −114.22 | 2058 | WUS_R | 2000–2021 | $-0.26\,[\pm0.80], p = 0.51$ |
| Great Smoky Mountains National Park | GSM | 35.66 | −83.61 | 1243 | EUS | 2000–2021 | $-4.51\,[\pm1.49], p < 0.01$ |
| Hohenpeißenberg | HPB | 47.80 | 11.01 | 985 | EUR | 2000–2021 | $-1.41\,[\pm1.00], p = 0.01$ |
| Izaña | IZO | 28.31 | −16.50 | 2373 | OT | 2000–2021 | $0.17\,[\pm0.84], p = 0.68$ |
| Joshua Tree National Park | JOT | 34.07 | −116.39 | 1244 | WUS_P | 2000–2021 | $-1.87\,[\pm1.57], p = 0.02$ |
| Jungfraujoch | JFJ | 46.55 | 7.99 | 3580 | EUR | 2000–2021 | $-0.70\,[\pm1.04], p = 0.18$ |
| Lassen Volcanic National Park | LAV | 40.54 | −121.58 | 1756 | WUS_R | 2000–2021 | $-1.28\,[\pm0.82], p < 0.01$ |
| Mauna Loa | MLO | 19.54 | −155.58 | 3397 | OT | 2000–2021 | $0.35\,[\pm1.45], p = 0.63$ |
| Mesa Verde National Park | MEV | 37.20 | −108.49 | 2170 | WUS_R | 2000–2021 | $-0.22\,[\pm1.01], p = 0.66$ |
| Monte Cimone | CMN | 44.19 | 10.70 | 2165 | EUR | 2000–2021 | $-2.47\,[\pm1.48], p < 0.01$ |
| Mount Bachelor Observatory | MBO | 43.98 | −121.69 | 2763 | WUS_R | 2004–2021 | $2.82\,[\pm2.23], p = 0.01$ |
| Mt. Kenya | MKN | −0.06 | 37.30 | 3678 | OT | 2002–2021 | $1.70\,[\pm2.45], p = 0.17$ |
| Petrified Forest National Park | PFN | 34.82 | −109.89 | 1712 | WUS_R | 2002–2021 | $-1.86\,[\pm0.91], p < 0.01$ |
| Pha Din | PDI | 21.57 | 103.52 | 1466 | OT | 2014–2021 | $-4.79\,[\pm11.75], p = 0.42$ |



| Pinedale | PND | 42.93 | −109.79 | 2388 | WUS_R | 2000–2021 | $-1.38\,[\pm0.74]$, $p < 0.01$ |
|---|---|---|---|---|---|---|---|
| Rangely | RAN | 40.09 | −108.76 | 1655 | WUS_R | 2010–2021 | $-2.30\,[\pm2.49]$, $p = 0.07$ |
| Rocky Mountain National Park | RMN | 40.28 | −105.54 | 2743 | WUS_P | 2000–2021 | $0.12\,[\pm0.81]$, $p = 0.77$ |
| Sequoia/Kings Canyon National Parks | SQA | 36.57 | −118.78 | 1890 | WUS_P | 2000–2021 | $-2.51\,[\pm1.84]$, $p = 0.01$ |
| Shenandoah National Park | SHN | 38.52 | −78.44 | 1073 | EUS | 2000–2021 | $-2.84\,[\pm2.03]$, $p = 0.01$ |
| Sonnblick | SNB | 47.05 | 12.96 | 3106 | EUR | 2000–2021 | $-1.41\,[\pm0.66]$, $p < 0.01$ |
| South Pole | SPO | −90.00 | −24.80 | 2841 | OT | 2000–2021 | $1.32\,[\pm0.41]$, $p < 0.01$ |
| Summit | SUM | 72.58 | −38.48 | 3238 | OT | 2000–2021 | $-2.37\,[\pm1.19]$, $p < 0.01$ |
| Whiteface Mountain | WFM | 44.37 | −73.90 | 1483 | EUS | 2000–2021 | $-2.00\,[\pm1.47]$, $p = 0.01$ |
| Wind Cave National Park | WNC | 43.56 | −103.48 | 1288 | WUS_R | 2005–2021 | $-0.67\,[\pm1.54]$, $p = 0.39$ |
| Yellowstone National Park | YEL | 44.56 | −110.40 | 2400 | WUS_R | 2000–2021 | $-1.54\,[\pm0.90]$, $p < 0.01$ |
| Yosemite National Park | YOS | 37.71 | −119.71 | 1599 | WUS_P | 2000–2021 | $-2.65\,[\pm1.68]$, $p < 0.01$ |
| Zion National Park | ZIO | 37.20 | −113.15 | 1213 | WUS_R | 2004–2021 | $-2.11\,[\pm0.97]$, $p < 0.01$ |
| Zugspitze | ZSF | 47.42 | 10.98 | 2671 | EUR | 2000–2021 | $-0.61\,[\pm0.68]$, $p = 0.13$ |

### 2.1.1 Surface ozone data selection

As our study focuses on the quantification of $O_3$ anomalies at high-elevation and remote locations, careful data selection was carried out, to focus on well-mixed atmospheric conditions, and to also avoid times of the day that can be influenced by fresh anthropogenic emissions that can lead to the localized production or destruction of $O_3$ (Cooper et al., 2020). As the sites in Fig. 1 are representative of very different environments, the analysis of the diurnal cycles led to the identification of the following conditions for data selection:

– Night-time (i.e., between 20:00 and 07:59 local time) data for mountaintop and stations above 1500 m a.s.l.. This selection was chosen to focus on regionally-representative $O_3$, and to avoid the presence of local air masses that are transported, during daytime, from the valleys up to the mountaintops by upslope winds (Price and Pales, 1963; Cooper et al., 2020; Cristofanelli et al., 2020). This condition was valid for all of the European sites above 1500 m a.s.l., for some of the OT sites (MLO, TLL, IZO, MKN and PDI), and MBO;

– Maximum daily 8-h average (MDA8) $O_3$ values for all of the US EPA and NPS sites (i.e., all stations belonging to WUS_R, WUS_P, and EUS, except MBO). This was chosen as these stations can experience surface deposition at nighttime, which can therefore lower the $O_3$ values. MDA8 values are typically characteristic of the time of the day when the boundary layer is well mixed, and are therefore representative of a broad region around each measurement site;

– Daily average data for HPB, SUM, and the two stations in Antarctica (DCC and SPO); the latter three sites are characterized by almost no diurnal $O_3$ cycle, and therefore all data from the full 24-h record can be used.



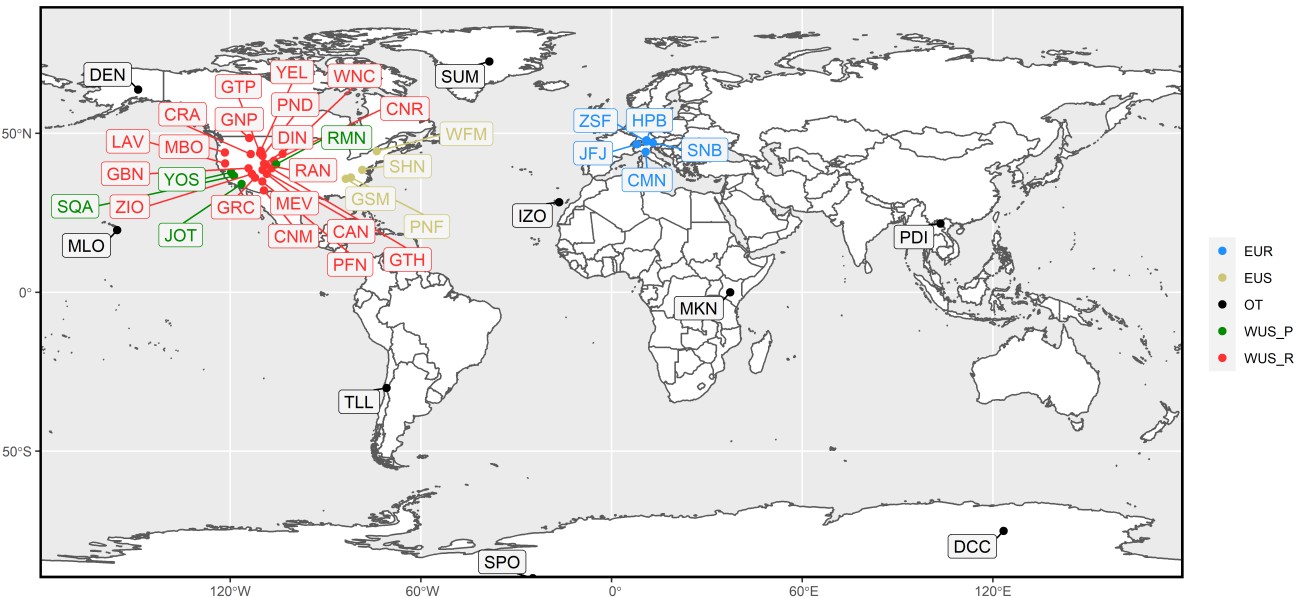

**Figure 1.** Geographical locations of the sites used in this study (details are reported in Table 1). The region abbreviations are as follows: WUS_R = Western US "rural", WUS_P = Western US "polluted", EUS = Eastern US, EUR = Europe, OT = Other.

### 2.1.2   Trend detection and calculation of the anomalies

To describe and quantify the effects of the COVID-19 economic downturn on $O_3$ values, we computed monthly $O_3$ anomalies

at each of the selected sites, derived after removing the seasonal and the trend components from the $O_3$ monthly averages. The deseasonalization allows to produce a more precise trend with less uncertainty, and avoid estimation bias due to missing data. Similar to Cooper et al. (2020), and Cristofanelli et al. (2020), we followed several steps to calculate the monthly $O_3$ anomalies.

First, we determined the monthly $O_3$ averages for each site, setting a threshold of 50% on hourly data availability for each

month. We also carried out a sensitivity study by adopting a different threshold (i.e., 66%) or by extending the threshold to daily averages before calculating monthly values. The sensitivity study produced no significant variation in our results (see Fig. S1 in the Supplementary Material). The choice of a more relaxed 50% threshold was made for retaining enough data at several "critical" sites, such as MKN, MBO, or PDI, which might suffer from issues that prevented complete data sampling in each month.

Second, we removed the seasonal cycle from the monthly averages, by calculating the month-by-month difference between the monthly averages and a "climatological year", composed of the 20-year mean for each of the 12 months. The baseline



period for the 20-year mean is 2000–2019; shorter periods were used if data availability is limited (see Table 1 for the different starting years).

Last, we used the differences calculated in the previous step for quantifying the long-term $O_3$ changes, and also to compute
the anomalies (i.e., deseasonalized and detrended) in order to further compare the consistency of the COVID-19 impact at different sites. We used quantile regression for evaluating the trends (and choosing the 50th percentile, i.e., equivalent to the median regression), which is recommended as a standard approach for trend analysis for the TOAR-II activity (Chang et al., 2023b). It is a well-suited technique for detecting heterogeneous distributional changes (Chang et al., 2021), and can incorporate covariates such as piecewise trends for change point analysis. Although this study places the focus on the impact
of COVID-19 economic downturn in 2020 and 2021, additional years of data will be required to determine if this event is a change point in the long-term trends. To account for autocorrelation and heteroscedasticity, the moving block bootstrap resampling algorithm is implemented (Lahiri, 2003): for each iteration the quantile regression model is fitted to a series of randomly selected block samples and the sampled trend value is extracted. The final trend value (and its uncertainty) was then determined by the mean (and standard deviation) of the sampled trend values. All trends are reported with their 95% confidence
interval and $p$-value.

## 2.2 IASI data

The IASI (Infrared Atmospheric Sounding Interferometer) instrument is a nadir-viewing Fourier transform spectrometer, flying on board the EUMETSAT (European Organisation for the Exploitation of Meteorological Satellites) Metop satellites (Clerbaux et al., 2009). The IASI instrument operates in the thermal infrared between 645 and 2760 cm$^{-1}$ with an apodized resolution
of 0.5 cm$^{-1}$. The field of view of the instrument is composed of a 2×2 matrix of pixels with a diameter at nadir of 12 km each. IASI scans the atmosphere with a swath width of 2200 km and crosses the equator at two fixed local solar times: 09:30 (descending mode) and 21:30 (ascending mode), allowing the monitoring of atmospheric composition twice a day at any location. Three versions of the instrument were built and launched at different times: one aboard the Metop-A platform (October 2006), one aboard the Metop-B platform (September 2012), and one aboard the Metop-C platform (November 2018).
Note that Metop-A was deorbited in October 2021.

Ozone profiles used to calculate $O_3$ partial columns for this study are described in Dufour et al. (2021). A data screening procedure is applied to filter cloudy scenes and to ensure the data quality (Eremenko et al., 2008; Dufour et al., 2010, 2012). It is worth noting that the maximum of sensitivity of the retrieved profile in the lower troposphere is around 4 to 6 km (Dufour et al., 2010, 2012). Therefore, we use the lower free tropospheric column product from 3 to 6 km. Only the morning overpasses
of IASI are considered in order to ensure a better sensitivity to the lower troposphere. To cover the longest possible period with consistent data, we consider only IASI on Metop-A in this study. A consistency analysis of the IASI-A, IASI-B and IASI-C time series is needed to use the three instruments simultaneously. Consistent with the surface $O_3$ measurements, we calculated anomalies for the $O_3$ partial columns, after removing the seasonality and the trend (see Sect. 2.1.2).





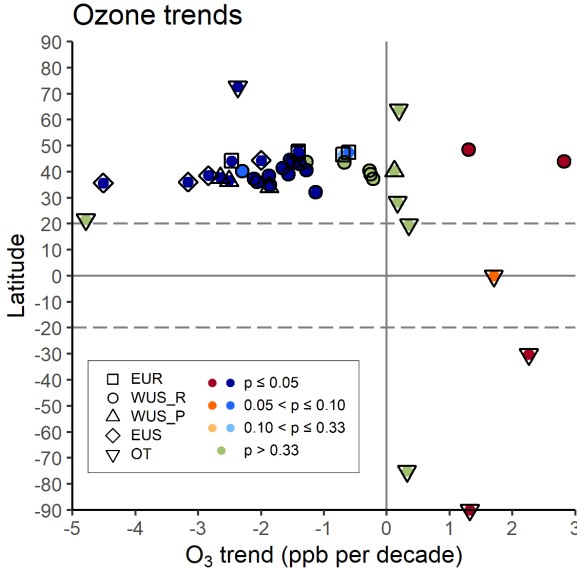

**Figure 2.** Decadal $O_3$ trends (50<sup>th</sup> percentile) for the 41 high-elevation sites used in this study. The reference periods for trend calculation for the different sites are listed in Table 1. Trends are ordered by latitude (y-axis), and the colors indicate the *p*-value on the trend. The shapes identify the different regions, i.e.: EUR = Europe, WUS_R = Western US "rural", WUS_P = Western US "polluted", EUS = Eastern US, OT = Other.

## 3    Results and discussion

### 3.1    Ozone trends

A trend analysis spanning the first two decades of the 21<sup>st</sup> Century for long-term observational datasets collected at high-elevation remote and rural locations was performed for most of the stations. The overall picture is reported in Fig. 2, where decadal $O_3$ trends are reported by latitude, and grouped by each of the regions considered in this study (Fig. 1). The calculation period of the trends is 2000–2021 (or shorter for some stations, when data back to 2000 were not available); a recent study (Chang et al., 2023a) has shown that, for MBO observatory, the long-term positive trend was clearly weakened when including the anomalous year 2020 compared to 2004–2019, but the trend rebounded in 2021, between the 5<sup>th</sup> and 95<sup>th</sup> percentiles. The variations of the long-term trends at all of the sites, computed by varying the calculation periods of the trends (i.e., 2000–2019, 2000–2020, and 2000–2021) are reported in Table S1. In several cases, the trends did not reveal any relevant impact of the 2020 anomalies, while for 10 sites the 2000–2020 trend was weakened compared to 2000–2019, and a rebound was observed when including 2021. It is interesting to note that for the European sites above 1500 m a.s.l. (i.e., CMN, JFJ, SNB, and ZSF) the long-term trend was weakened when including 2020, and continued to weaken with the addition of 2021 data (see Table S1).





Considering the full 2000–2021 record, the observed trends for the 41 high-elevation sites vary greatly, from −4.79 to 2.82 ppb per decade. Decreases in surface $O_3$ were observed for 30 European and North American sites, with the exceptions of MBO (2.82 ppb per decade) and GNP (1.30 ppb per decade). The trend values for the sites belonging to the OT category showed large differences, with $O_3$ increases recorded for TLL (2.26 ppb per decade) and MKN (1.70 ppb per decade); on the other hand, PDI (although limited by the rather short reference period) and SUM station showed $O_3$ decreases (−4.79 and −2.37 ppb per decade, respectively). Both Antarctic sites showed positive trends (SPO: 1.33 ppb per decade, and DCC: 0.32 ppb per decade), which are in line with previous studies described in Kumar et al. (2021). It has to be noted that the trend at DCC could be affected by the large data gap in the measurements between 2014 and 2016, and this will certainly need further investigation.

The trends for most of the sites in Western North America considered in this study were previously reported by Chang et al. (2023a), although considering a slightly longer period (1995–2021). Their results are consistent with the ones reported in Fig. 2 for WUS_R and WUS_P categories, indicating that the majority of the sites in Western North America show a consistent pattern of negative trends, pointing to an overall decrease of regional boundary layer $O_3$. The clear outlier is MBO, but this site is uniquely situated on the summit of an isolated mountain. During nighttime conditions reported here, MBO is strongly influenced by the lower free troposphere which has experienced a small increase of $O_3$ since the 1990s (Chang et al., 2023a); during summer and autumn MBO is also impacted by ozone produced from western forest fires, which have become more frequent in recent years (Farley et al., 2022; Jaffe et al., 2022). It has to be noted that the inclusion of 2020 and 2021 in the analysis did not cause any notable variation in the trend values for several sites across the Western US (see Table S1), with the effects of the COVID-19 economic downturn on the long-term trends only visible when considering the spring season (Chang et al., 2023a).

Despite their variability when considering different periods for the trend calculation, the trends for the European sites over 2000–2021 showed persistent negative values when compared to previous literature (e.g., Cristofanelli et al., 2020; Christiansen et al., 2022). While the trends for JFJ and SNB remained almost unchanged, CMN showed a larger negative trend with respect to the 1996–2016 trends reported in Cristofanelli et al. (2020), and ZSF showed a higher (i.e., a less negative) value with respect to this reference period. The positive trends in the Southern Hemisphere are in line with the modeled trends reported by Wang et al. (2022), and with the trends obtained from the TCR-2 chemical reanalysis (Miyazaki et al., 2020).

Regarding positive trends, model studies report increases in the tropospheric ozone burden occurring mainly in the free troposphere (700–250 hPa, see Fiore et al., 2022), while the surface trends tend to be mixed, especially for the extratropical regions in the Northern Hemisphere (see also Miyazaki et al., 2020; Chang et al., 2023a). This is indeed the case for the European and North American sites reported here (see Fig. 2), indicating that surface $O_3$ trends are often not related to the trends observed in the free troposphere (Gulev et al., 2021), as also reported by Chang et al. (2023a). However, we emphasize that the sites in Fig. 2 only cover a limited portion of the Earth's surface, as we are limited by the available observations, and these results cannot be assumed to be representative of the entire World.





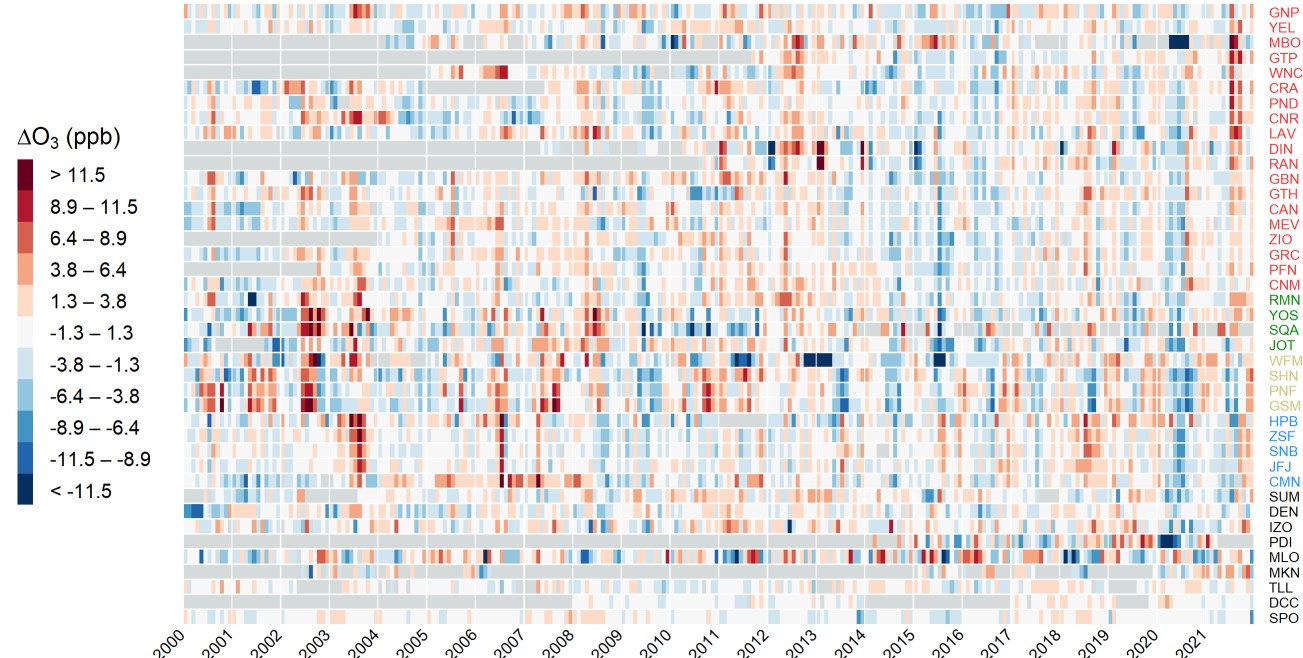

**Figure 3.** Heatmap of the monthly $O_3$ anomalies ($\Delta O_3$) for the sites used in this study. The sites are grouped by region (the different colors identify the regions, see Fig. 1), and ordered by decreasing latitude.

## 3.2 Quantification of the anomalies

Figure 3 provides a detailed summary of the anomalies for each site, which are grouped by region and ordered by latitude. Figure S2 in the Supplementary Material shows the same anomalies, but in the form of monthly time series, together with the average anomalies for the different regions. Figure 3 clearly shows widespread persistent negative anomalies affecting most of the sites in 2020, both in spring (March–May, i.e., MAM) and summer (June–August, i.e., JJA). The situation was somewhat similar in 2021, although some sites showed partial $O_3$ rebounds (e.g., the sites in the Western US). A closer look at the average seasonal differences for the regions is provided in Table 2, while the focus on the spatial distribution of the anomalies (for the sites in North America and Western Europe) for 2019, 2020, and 2021 is provided in Fig. S3–S5 of the Supplementary Material.

By analyzing seasonal averages of the anomalies, the Western US rural (WUS_R) sites experienced persistent negative anomalies for MAM and JJA 2020 ($-6\%$ and $-5\%$, respectively), and for MAM 2021 ($-2\%$), while JJA 2021 was characterized by a strong positive anomaly (2.8 ppb, 6%). Late summer 2020 was characterized by the spread of wildfires in the Western US (Filonchyk et al., 2022; Jaffe et al., 2022; Peischl et al., 2023; Langford et al., 2023), resulting in positive average anomalies for August and September 2020 (2.3 and 2.5 ppb, i.e., 5% and 6%, respectively) for several sites (see Fig. S4). Without considering August 2020, the JJA 2020 seasonal average would result in a much more pronounced negative anomaly, i.e.,





**Table 2.** Seasonal average anomalies for the different regions considered in this study. Values in brackets indicate percentage variations. The region abbreviations are as follows: WUS_R = Western US "rural", WUS_P = Western US "polluted", EUS = Eastern US, EUR = Europe, OT = Other.

| Season | WUS_R | WUS_P | EUS | EUR | OT |
|---|---|---|---|---|---|
| MAM 2020 | −3.1 ppb (−6%) | −4.3 ppb (−9%) | −2.5 ppb (−4%) | −2.1 ppb (−3%) | −2.1 ppb (−3%) |
| JJA 2020 | −2.3 ppb (−5%) | −2.9 ppb (−4%) | −5.6 ppb (−12%) | −4.8 ppb (−8%) | −1.5 ppb (−5%) |
| MAM 2021 | −0.9 ppb (−2%) | 0.4 ppb (3%) | 0.9 ppb (3%) | −2.7 ppb (−4%) | −1.3 ppb (−1%) |
| JJA 2021 | 2.7 ppb (6%) | 1.2 ppb (3%) | −3.8 ppb (−8%) | −2.6 ppb (−5%) | 0.4 ppb (1%) |

−4.5 ppb (−10%), giving an indication of the magnitude of the secondary production of $O_3$ following the spread of wildfires, and thus partly influencing the strong negative anomaly that characterized this region following the 2020 COVID-19 economic downturn. As reported by World Meteorological Organization (2021), the fire season in Western North America in 2021 was also very intense, with the annual total estimated emissions ranking in the top five years of 2003–2021, and contributed to widespread air pollution. The emissions produced by the large widespread wildfires that impacted North America in these months can also explain the different patterns in the Western US compared to Eastern US and Europe (see Fig. 4).

The situation for the four Western US sites downwind of polluted areas (WUS_P) was slightly different, with the negative anomalies being larger than those of WUS_R in MAM 2020 (−4.1 ppb, −9%), and positive anomalies for 2021 (3% for both MAM and JJA 2021). The average anomaly in July 2021 is slightly weaker compared to the WUS_R average, due to the negative anomaly at the JOT site (Fig. 4), despite the other stations being heavily impacted by the North American wildfires (the average, excluding JOT, for JJA 2021 was 2.8 ppb, 5%).

The sites in the Eastern US (EUS) category experienced negative anomalies in 2020 (−4% and −12% for MAM and JJA, respectively) and in JJA 2021 (−8%), and a positive anomaly in MAM 2021 (3%). It is interesting to note that, in both of the summer seasons, the EUS sites exhibited an opposite structure with respect to WUS_R and WUS_P sites.

The European sites (EUR) were characterized by persistent negative anomalies throughout all of the considered seasons in Table 2. MAM 2020 reported a total negative anomaly (−2.0 ppb, −3%), but was characterized by an interesting "bump" in $O_3$ concentrations in April, with values almost comparable to the 2000–2019 values, for all stations (even HPB at lower elevation registered a positive anomaly for April 2020, see Fig. 3 and Fig. S4 in the Supplementary Material). This feature was previously observed at CMN by Cristofanelli et al. (2021), who reported that these higher $O_3$ values were possibly attributed to the frequent occurrence of transport from the free troposphere, or from areas usually not considered as sources of anthropogenic pollution (i.e., the Mediterranean Sea or northern Africa), or to the transport of stratospheric air masses. The negative anomalies then continued (except a positive "bump" in June 2021 for HPB, SNB, and ZSF) until September 2021, when all EUR sites experienced a rebound in $O_3$ values, and registered positive anomalies until the end of the year.

While Table 2 reports average values for the "Other" (OT) sites, a consistent "global" picture cannot be drawn, as these sites behaved very differently from each other (see Fig. 3). The SUM (and, partly, IZO) anomalies are more in line with the



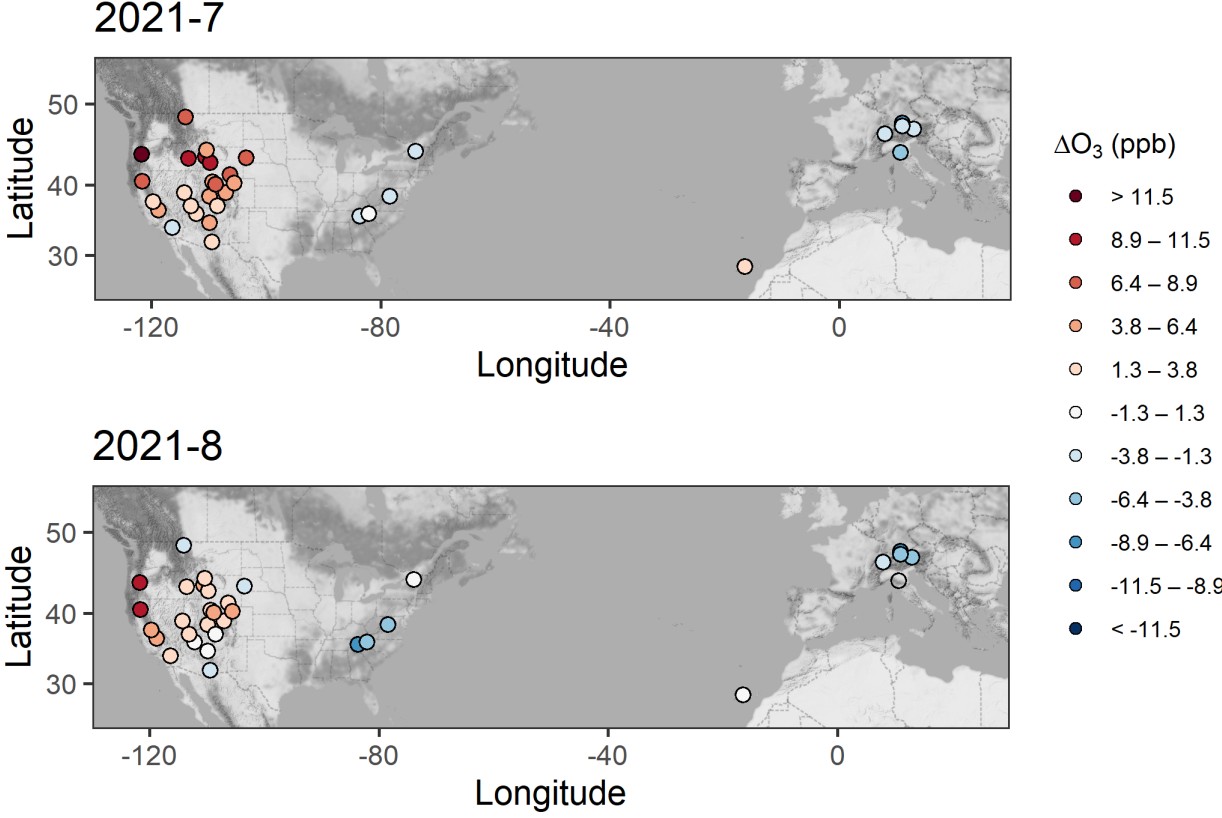

**Figure 4.** Spatial distribution of the anomalies ($\Delta O_3$) for July (top) and August (bottom) 2021, for sites in North America and Western Europe. The full series of monthly maps for 2019, 2020, and 2021 is provided in the Supplementary Material (Fig. S3–S5).

EUR sites, while DEN, MLO, MKN, and TLL had alternating positive and negative anomalies. PDI showed by far the largest negative anomalies in the first half of 2020 (average of $-8.3$ ppb from January to October, $-20\%$), but unfortunately no information on possible $O_3$ recovery in 2021 is available due to missing data. The distant Antarctic sites, on the other hand, did not reveal any signal of influence from the COVID-19 economic downturn, with $O_3$ values perfectly in line or even higher
than the climatological averages for both DCC and SPO. For more details about the interannual variability at each site, please refer to Fig. S6–S46 in the Supplementary Material.

### 3.3 Anomaly attribution

The results presented in Sect. 3.2 are in line with those reported in Ziemke et al. (2022), who observed reduced values of tropospheric column ozone (TCO) in spring–summer 2020 and 2021, and who attributed the decrease to reduced pollution (i.e.,
reductions of $\sim$10–20% in tropospheric $NO_2$ in the Northern Hemisphere). More specifically, Ziemke et al. (2022) indicate a reduction of 3 dobson units of TCO, corresponding to a $\sim$7–8% decrease for the area 20° N–60° N. If we consider the seasonal





averages, excluding the OT category, we obtain almost comparable results for 2020 also for the surface $O_3$ observations (average negative anomalies of $-6\%$ and $-7\%$ for MAM and JJA, respectively). The situation is different in 2021, where we obtain higher values ($0\%$ for MAM, and $-6\%$ for JJA if we consider EUS and EUR only, to exclude the wildfires' influence).

However, it has to be noted that we considered only a selection of sites, and that in some cases our seasonal averages can be determined by a combination of sub-seasonal positive and negative anomalies (see Sect. 3.2), possibly due to the impact of other "local" factors.

### 3.3.1 Column $O_3$ variability from IASI

Reductions in mid-tropospheric $O_3$ seen by the IASI satellite instrument are similar to the ones from the surface observations.

Figure 5 shows the 2008–2020 variability of $O_3$ in the 3–6 km column (both monthly averages and anomalies), for three specific regions: (i) an area around the European Alpine sites (i.e., EUR, 40–50° N, 5–20° E), (ii) the Eastern US (EUS, 35–50° N, 85–70° W), and (iii) the Western US (WUS, 30–50° N, 125–100° W). As stated in Sect. 2.2, the 3–6 km column corresponds to the maximum sensitivity of the IASI retrieval in the free troposphere and is, thus, superior to the 0–3 km column, where the retrieval sensitivity is more limited and the column is not independent of the column above. We did not include 2021 in the

analysis as IASI-A operations stopped before the end of the year and all the measurements were not done in the nominal mode of the satellite and the instrument.

In all three regions reductions in the 3–6 km column $O_3$ were observed in 2020, both for the monthly averages and the anomalies. Average negative anomalies were continuously observed throughout MAM and JJA 2020. The anomalies in MAM 2020 were quite similar among the regions, i.e., $-3\%$ for EUS, $-4\%$ for EUR, and $-5\%$ for WUS. A similar anomaly ($-6\%$)

was observed for WUS also in JJA 2020, and these negative values persisted also in fall (SON), indicating that the wildfire influence had only a minor impact at this upper layer with respect to the surface monitoring sites. For both EUR and EUS, negative anomalies were still observed in JJA 2020 ($-1\%$ and $-7\%$, respectively), and a rebound occurred in SON, with values falling within 1 standard deviation from the 2008–2019 climatological average. The smaller EUR anomaly with respect to EUS in JJA 2020 can be explained by the "bump" that characterized the European region in June.

Despite the differences due to the subsets investigated in this study, these results are comparable to the reductions in free tropospheric $O_3$ observed by Steinbrecht et al. (2021), i.e., $-7\%$ (with respect to the 2000–2020 climatological mean) from April to August and for the 1–8 km layer in the Northern Hemisphere. Moreover, the behavior of the anomalies observed here is consistent with the tropospheric $O_3$ anomalies shown by Miyazaki et al. (2021) and Ziemke et al. (2022) discussed above, including the rebound in SON resulting in column $O_3$ values comparable to the previous years. However, it has to be noted

that our anomalies are weaker than the ones presented in Ziemke et al. (2022), as we are limiting the IASI measurements to land regions around our measurement sites, while Ziemke et al. (2022) observed the largest negative 2020 and 2021 anomalies above the ocean areas of the Northern Hemisphere.





**Figure 5.** Annual variability of the 3–6 km column O$_3$ monthly averages (top row) and anomalies (bottom row) from IASI, for the three regions considered (i.e., EUR, EUS and WUS, for details on definitions refer to Sect. 3.3.1). The gray lines indicate the single years from 2008 to 2019, the black line is the 2008–2019 climatology (together with ±1 standard deviation, dotted lines), and the red line indicates 2020.

### 3.3.2 Emissions reductions

To investigate the reductions in the emissions, we analyzed data from the Carbon Monitor, a near-real-time dataset of global 275 CO$_2$ emissions from fossil fuels and cement production, available since January 2019 (Liu et al., 2020).

Table 3 reports the CO$_2$ global emissions variations from Carbon Monitor, divided into the different sectors, for the combinations of 2019, 2020, and 2021. As also done above for the characterization of the anomalies, here we consider 2019 as the reference year for "pre-COVID-19" emissions, and 2020 and 2021 as being the ones affected by the COVID-19 economic downturn, and with a possible recovery in emissions. By analyzing all sectors together, we can immediately spot the decrease



**Table 3.** $CO_2$ global emissions variations (expressed in %) from Carbon Monitor (Liu et al., 2020), for the different combinations of years 2019, 2020, and 2021, and with focus on MAM and JJA for each comparison. The percentage represents the contribution of each sector to the total change (i.e., "All sectors"), while the percentage in parentheses indicates the sector change in the selected year with respect to the comparison year.

| Sector | 2020 vs 2019 | | | 2021 vs 2019 | | | 2021 vs 2020 | | |
|---|---|---|---|---|---|---|---|---|---|
| | All | MAM | JJA | All | MAM | JJA | All | MAM | JJA |
| All sectors | −5.3% | −13.6% | −4.0% | +0.5% | +0.9% | +1.5% | +6.1% | +16.8% | +5.7% |
| Power | −1.1% | −3.3% | +0.0% | +1.5% | +1.6% | +2.7% | +2.7% | +5.7% | +2.9% |
| | (−2.8%) | (−9.0%) | (−0.1%) | (3.9%) | (4.3%) | (6.7%) | (6.9%) | (14.7%) | (6.8%) |
| Industry | −0.7% | −3.3% | −0.8% | +0.7% | +1.3% | +0.4% | +1.5% | +5.4% | +1.2% |
| | (−2.5%) | (−10.6%) | (−2.4%) | (2.2%) | (4.2%) | (1.1%) | (4.9%) | (16.5%) | (3.6%) |
| Ground transport | −2.0% | −5.0% | −1.4% | −0.6% | −0.8% | −0.5% | +1.5% | +4.8% | +0.9% |
| | (−10.9%) | (−26.1%) | (−7.3%) | (−3.1%) | (−4.2%) | (−2.7%) | (8.8%) | (29.6%) | (4.9%) |
| Residential | −0.2% | −0.3% | +0.1% | −0.1% | −0.1% | +0.0% | +0.0% | +0.2% | +0.0% |
| | (−1.6%) | (−2.9%) | (1.0%) | (−1.2%) | (−0.9%) | (0.2%) | (0.4%) | (2.1%) | (−0.8%) |
| Domestic aviation | −0.3% | −0.5% | −0.4% | −0.1% | −0.1% | −0.1% | +0.2% | +0.4% | +0.3% |
| | (−30.8%) | (−49.6%) | (−38.0%) | (−13.1%) | (−11.9%) | (−10.3%) | (25.5%) | (74.7%) | (44.5%) |
| International aviation | −1.0% | −1.2% | −1.5% | −0.9% | −1.0% | −1.0% | +0.1% | +0.2% | +0.5% |
| | (−56.0%) | (−67.0%) | (−71.2%) | (−48.2%) | (−58.5%) | (−48.0%) | (17.7%) | (25.5%) | (80.2%) |

in emissions occurring in 2020 with respect to 2019 (−5.3%), and the strong rebound of emissions in 2021 (+0.5% and +6.1% with respect to 2019 and 2020, respectively). The rebound of emissions near "pre-COVID-19" levels was also analyzed in other recent works, indicating that fossil fuel $CO_2$ emissions in 2021 nearly pushed global emissions back to 2019 levels (Jackson et al., 2022), and that 2021 emissions would have even exceeded the 2019 values if not for several low-income countries that had not recovered from the pandemic yet (Davis et al., 2022).

As the strongest $O_3$ anomalies presented in this study are clustered in the Western US and Europe, we also focused on regional $CO_2$ anomalies, by analyzing the US and Europe values provided by Carbon Monitor (see Table S2 and S3 in the Supplementary Material). In this case, no distinction between the Western and Eastern US is made, and Europe is considered as composed of the emissions in the 27 European Union countries plus the United Kingdom. While the decrease in 2020 emissions with respect to 2019 was evident for both regions (−10.9% and −10.1% for Europe and US, respectively), the

rebound to "pre-COVID-19" levels (i.e., 2021 against 2019 emissions) was smaller for these two areas, with respect to the global rebound (−2.7% and −4.5% for Europe and US, respectively). This may be one of the causes for the persistent $O_3$ negative anomalies that still characterized 2021. For Europe, the $O_3$ negative anomalies that were observed throughout MAM and JJA 2021 could be partly explained by an incomplete recovery in the emissions (−2.9% and −6.3% for MAM and JJA,





respectively, considering all sectors together). For the US, the $CO_2$ anomalies are more evident in MAM 2021 ($-5.8\%$) than
in JJA ($-0.3\%$).

Analyzing the different sectors separately, the limits imposed on domestic and international aviation caused the largest
negative variations in 2020 with respect to "pre-COVID-19" levels; these sectors witnessed the largest rebounds in 2021,
although not returning to 2019 levels (and this was particularly true for international aviation, where a total difference of
$-48.3\%$ and $-33.9\%$ was still observed, for Europe and US, respectively). Particularly for the rural and remote sites, the
aircraft emissions play a key role in determining the tropospheric $O_3$ trends, mainly because of the aircraft emitting NOx in
the mid- and upper-troposphere, where the $O_3$ production efficiency is high (Wang et al., 2022). Therefore, this incomplete
recovery in aircraft emissions for 2021 could partly explain the persistent negative anomalies observed. Also ground transport
and, to a lesser extent, residential emissions variations showed the same behavior (while this was true for the global and US
emissions, Europe had larger emissions for these two sectors in 2021 with respect to 2019, see Table S2). On a global scale,
rebounds in 2021 for power and industry were so large that the emissions in this year exceeded those of 2019 (3.9% and 2.2%
for power and industry, respectively); on the other hand, positive emissions anomalies in these two sectors were observed for
European industry emissions only, with negative emissions observed in the US and for the power sector for both regions.

### 3.4 Possible $O_3$ recovery in 2022

The data presented in this study concerned the first full year after the 2020 COVID-19 economic downturn (i.e., ending in
December 2021), therefore little information on the possible recovery of $O_3$ values to "pre-COVID-19" levels is present.
Nevertheless, the datasets discussed here are to date the most comprehensive time series for investigating these anomalies
from high-elevation stations. The availability of validated 2022 data for the four mountaintop WMO/GAW global stations in
Europe (i.e., CMN, SNB, ZSF, and JFJ) allowed us to investigate the possible rebound of $O_3$ values for this specific European
area that encompasses the Alps and the northern Apennines (see Fig. 6). In this case, the monthly data were again detrended
before the calculation of the anomalies, but with respect to the whole 2000–2022 period; the reference for the calculation of
the climatology was still the 2000–2019 period.

At all four sites, the negative 2000–2019 trends became increasingly more negative when including the 2020 and 2021 data
(see Sect. 3.1 and Table S1). But the inclusion of the 2022 data slightly shifted the trends back towards the pre-pandemic
levels, i.e.: $-2.35$ ppb per decade ($\pm1.53$ ppb per decade, $p < 0.01$) for CMN, $-1.34$ ppb per decade ($\pm0.74$ ppb per decade,
$p < 0.01$) for SNB, $-0.52$ ppb per decade ($\pm0.77$ ppb per decade, $p = 0.18$) for ZSF, and $-0.64$ ppb per decade ($\pm1.04$,
$p = 0.22$) for JFJ.

When looking at monthly $O_3$ values and anomalies (Fig. 6), an overall rebound for 2022 seems evident in the first part of
the year (January to March) for all of the four sites, with monthly averages comparable to the climatology, while the anomalies
from April to June showed negative values. The values for the rest of the year were generally within one standard deviation
from the climatological averages, with two months (July and August) exhibiting higher monthly averages with respect to the
2000–2019 baseline. The characteristics of the $O_3$ rebound in 2022, which are commonly shared among the high-elevation sites
located in Western Europe, will certainly need deeper investigation, especially for the attribution of the lower values observed





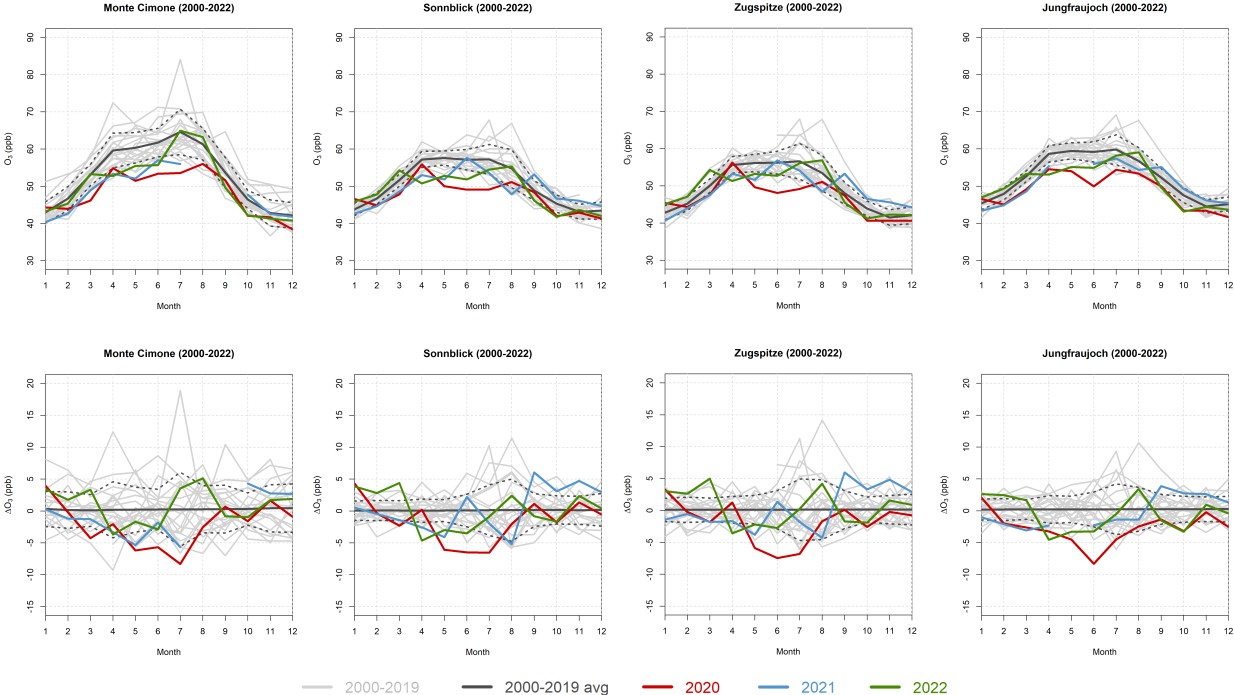

**Figure 6.** Annual variability of the $O_3$ monthly averages (top row) and anomalies (bottom row) at CMN, SNB, ZSF, and JFJ. The gray lines indicate the single years from 2000 to 2019, the black line is the 2000–2019 climatology (together with $\pm 1$ standard deviation, dotted lines), and the red, blue, and green lines indicate 2020, 2021, and 2022, respectively.

from April to June, given that no restrictions driving the variability of the $O_3$ precursors were present in 2022. Other than meteorological variations, mineral dust transport has been proven to significantly reduce the $O_3$ values at these high-elevation sites (e.g., Duchi et al., 2016). As the first half of 2022 was largely affected by Saharan mineral dust transport events reaching Western Europe (both in March and June 2022), these could have played an important role in lowering the $O_3$ values in this period.

## 4 Conclusions

In this paper we demonstrated that the negative $O_3$ anomalies that were observed in the free troposphere in recent studies also occurred in the boundary layer surrounding several high-elevation sites. This was performed by investigating the surface $O_3$ variability at 41 high-elevation sites regionally distributed, following the COVID-19 economic downturn that occurred in 2020 and the following year, 2021, associated with a recovery of emissions. Widespread persistent negative anomalies were observed both in spring (MAM) and summer (JJA) 2020 for all of the regions considered in this study, while for 2021 continuous negative anomalies throughout MAM and JJA were observed only for Europe and, partially, for the Eastern US. On the other hand, the



Western US sites were heavily impacted by wildfire emissions in 2021, resulting in positive anomalies, especially for JJA and for the rural sites. A global picture for the rest of the World could not be drawn, as the sites were spanning a range of different environments and did not show consistent patterns.

The anomaly behavior was further studied by analyzing the variability in the column $O_3$ from the IASI satellite products. Consistent with previous studies (e.g., Miyazaki et al., 2021; Ziemke et al., 2022), negative anomalies were observed also in
the free-tropospheric 3–6 km column $O_3$ product, for both MAM and JJA 2020 (−4% for both seasons, on average over the considered regions). These results indicate that one of the causes of such widespread anomalies is the reduction in the emissions of the $O_3$ precursors. To further assess this point, we also investigated the reduction in the emissions for the different sectors for the years 2019, 2020, and 2021, as derived from Carbon Monitor, a near-real-time dataset of global $CO_2$ emissions. The results highlight the decrease in emissions that occurred in 2020 with respect to 2019 (−10.9% and −10.1% analyzing all sectors
together, for both Europe and US, respectively), and the rebound of emissions in 2021 that took place globally. However, the recovery in emissions in 2021 did not reach "pre-COVID-19" levels of 2019 in the two macro-regions that encompass most of the sites investigated here (−2.7% and −4.5% for Europe and US, respectively), and this could be one of the causes for the persistent negative anomalies that were observed in these two areas.

As our dataset was limited to the first full year after the 2020 COVID-19 economic downturn, few conclusions could be drawn
regarding the full recovery of $O_3$ values to "pre-COVID-19" levels. However, we made use of 2022 data for four mountaintop sites in Western Europe, and we observed a common pattern concerning $O_3$ variability in 2022. This was characterized by a rebound in the first part of the year (January to March), with monthly values comparable to the 2000–2019 climatology; then, from April to June negative anomalies were observed, and the values for the remaining part of the year were within one standard deviation from the climatological averages. The rebound in $O_3$ values starting from 2021–2022, will certainly
need deeper investigation, especially concerning the attribution of the wide-ranging variability, and will be the matter of future research.

*Data availability.* The ozone data for CMN, DCC, HPB, IZO, JFJ, MLO, MKN, PDI, SNB, SPO, SUM, TLL, and ZSF stations can be retrieved from the WMO/GAW World Data Center for Reactive Gases (WDCRG) hosted by NILU (https://ebas.nilu.no/). Data for MBO are permanently archived by the University of Washington at its ResearchWorks archive, see https://sites.uw.edu/jaffe-group/mt-bachelor-
observatory/. The Clean Air Status and Trends Network (CASTNET) ozone data can be retrieved at https://www.epa.gov/castnet. Carbon Monitor data were downloaded from https://carbonmonitor.org/ and the data presented in this study refer to the April 30th, 2023 data update. The IASI dataset is currently under preparation for repository deposition and the corresponding DOI will be provided in the final version of the manuscript.

*Author contributions.* DP, PC, and ORC contributed to the conception and design of the study. DP conducted the data analysis with inputs
from ORC, PC, and KLC. GD contributed the IASI data and advised on their interpretation. GB, CC, PE, DJ, DK, JL, IP, MP, TS, BCS, MS,



CT, and PC provided the ozone measurements. DP drafted the paper with inputs from ORC, PC, KLC, and GD. All authors contributed to the discussion and improvement of the paper.

*Competing interests.* ORC is the Scientific Coordinator of the TOAR-II Community Special Issue, to which this paper has been submitted, but he is not involved with the anonymous peer-review process of this or any of the other papers submitted to the Special Issue journals.

*Acknowledgements.* We would like to thank Iris Buxbaum and Wolfgang Spangl, Umweltbundesamt/Federal Environment Agency, Austria, for providing the Sonnblick data; the Dirección Meteorológica de Chile for providing data from El Tololo, Chile; the Kenya Meteorological Department for providing data from Mt. Kenya, Kenya; the Vietnam National Centre for Hydro-Meteorological Forecasting for providing data from Pha Din, Vietnam; Robert Holla, Deutscher Wetterdienst/German Meteorological Service, for providing data from Hohenpeißenberg, Germany. The research leading to these results has received funding from the European Union's Horizon 2020 research and innovation programme under grant agreement No 654109. The ozone measurements at Concordia, Antarctica, were carried out during several Italian Antarctic Research Programme (PNRA) projects, such as "LTCPAA – Long-term Measurements of Chemical and Physical Properties of Atmospheric Aerosol at Dome C" and "STEAR – Stratosphere-to-Troposphere Exchange in the Antarctic Region", and the authors thank the joint French-Italian Concordia Program and the logistics team (IPEV-PNRA) for their kind assistance during the experimental campaigns. Owen R. Cooper and Kai-Lan Chang were supported by NOAA cooperative agreement NA22OAR4320151.





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
