# Peer review of "Fingerprints of the COVID-19 economic downturn and recovery on ozone anomalies at high-elevation sites in North America and Western Europe"

_EGUsphere, 2023_

## Community Comment (CC1)

Comments by Rodrigo J. Seguel on behalf of the TOAR-II Steering Committee on:

**Fingerprints of the COVID-19 economic downturn and recovery on ozone anomalies at high-elevation sites in North America and Western Europe**

Davide Putero (corresponding author), Paolo Cristofanelli, Kai-Lan Chang, Gaëlle Dufour, Gregory Beachley, Cédric Couret, Peter Effertz, Daniel A. Jaffe, Dagmar Kubistin, Jason Lynch, Irina Petropavlovskikh, Melissa Puchalski, Timothy Sharac, Barkley C. Sive, Martin Steinbacher, Carlos Torres, and Owen R. Cooper

This manuscript was submitted to ACP as part of the TOAR-II Community Special Issue https://doi.org/10.5194/egusphere-2023-1737
Discussion started: 16 August 2023; discussion closes 27 September 2023

This review is by Rodrigo Seguel, member of the TOAR-II Steering Committee. The primary purpose of these reviews is to identify any discrepancies across the TOAR-II submissions, and to allow the author teams time to address the discrepancies. Additional comments may be included with the reviews.

**General comments**

The manuscript addresses the impact of the COVID-19 pandemic on the ozone mixing ratio measured at 41 high-altitude surface monitoring stations, mainly in the United States and Western Europe, during 2020 (downturn) and 2021, as the beginning of the global economic recovery. The paper adopts the guidelines suggested by TOAR-II. In this regard, the authors applied quantile regression to estimate trends (Chang et al., 2023), facilitating future comparisons with ongoing papers to be summited in this special issue. The study also analyzes the variability in the ozone column (3-6 km) from the IASI satellite products. Surface negative anomalies, especially in 2020, are consistent with IASI observations and previous publications (Ziemke et al., 2022). Additionally, the study relates wildfires to positive anomalies observed in the Western US. Therefore, the paper is a valuable contribution to the TOAR-II Community Special Issue.

**Minor comments**

**Table 1**: The Denali National Park (DEN) station is not exactly a high-elevation site (663 m a.s.l.). However, there is no explicit definition to classify high-elevation sites provided, at least by TOAR-II. Can the authors indicate or clarify the motivation to include this station?

**Line 33**: "non-methane volatile organic carbons (NM-VOCs)". Please substitute "carbons" by "compounds," which is the standard definition. Alternatively, one finds in the literature non-methane hydrocarbons (NMHC), which is not accurate because do not involve other heteroatoms present in the chemical structures, such as oxygen and nitrogen, among others.

**Line 104, 108**: I assume that most of the 41 stations meet the 75% threshold. Is it possible, for instance, to indicate those stations with data availability lower than 75% in Table 1?

To what extent are the MKN, MBO, or PDI critical to the analysis? In this regard, I suggest rewording the sentence: "which might suffer from issues that prevented complete data sampling in each month."

**Line 208**: Can the authors check the reference WMO (2021)? WMO 2021 describes the intense wildfire season of 2020, not "Western North America in 2021".

**Line 221, 228, 264**: The term "bump" is a positive anomaly probably due to transport processes, as stated by the authors. I suggest not using "bump" because it is unclear and can be described using standard terms.

---

## Author Comment (AC1)

**Reviewer #1:**

*General comments*

This paper investigates the fingerprints of Covid-19 on 41 elevated mountain sites over the world, mainly in the USA and Europe. The scientific interest of the paper is very important, regarding the ozone chemistry related to sources and sinks. The paper is excellently written and well organised with the different chapters.

The measurements sites and the methods for data selection are well described and the data selection is accurate, with night-time values or daily 8h maximum averages for some stations. The use of IASI data is a good choice for comparing with satellite data.

The quantification of the 2020-2021 anomalies is well explained and discussed, related to the emissions reductions shown in Table3.

The conclusion is robust, due to the high number of sites and the O3 reduction is comparable to the IASI data.

All figures are excellent quality, easily understandable and well commented in the text. The supplementary material is also excellent quality.

This paper is suitable for publication.

*We thank the reviewer for his/her/their positive review and encouraging evaluation. In the following, we report our point-to-point replies to each of the raised points. Modifications to the text are performed in the revised version of the manuscript and are marked in different colors.*

*Minor comments*

The author should very briefly discuss about a possible reduction of stratosphere/troposphere exchanges in the period, as a non-negligible part of free troposphere ozone is coming from the stratosphere and as the stations are located in altitude.

*We thank the reviewer for the feedback. 2020 was an interesting year due to an anomalous $O_3$ depletion event in the Arctic stratosphere, which has been discussed in several works (e.g., Dunn et al., 2021; Steinbrecht et al., 2021). In particular, Fig. 2.57 of Dunn et al. (2021) shows the entire evolution of the event, with the total column of $O_3$ over the Northern Hemisphere showing a minimum from February to April/May. By June, the total column of $O_3$ rapidly recovered towards its climatological values. As also shown in Chang et al. (2022), in June there was no clear impact of this depletion event on the free tropospheric $O_3$ anomalies. Steinbrecht et al. (2021), by using simulations from the NASA GMI model, observed that the depletion event in 2020 contributed to less than one quarter on the observed anomalies in the troposphere. Moreover, Ziemke et al. (2022) indicate that the observed reduction in stratosphere-to-troposphere exchange (STE) in 2020 did not drive the anomalies in the free troposphere, as the satellite measurements showed negative tropospheric $O_3$ anomalies in both 2020 and 2021, whereas the meteorological conditions controlling the strength of STE were close to the climatological means in 2021; thus, Ziemke et al. (2022) suggested that the tropospheric anomalies can be largely attributed to decreases in emissions. This was further confirmed by our results: Table 2 shows that most of the regions considered in this study showed the largest $O_3$ anomalies in JJA 2020 rather than MAM 2020. Therefore, we hypothesize that the reduction in the*

*stratosphere-to-troposphere transport occurred in 2020 could have played only a minor role in modulating the $O_3$ anomalies. A sentence was added in Sect. 3.3.*

Chang, K.-L., Cooper, O. R., Gaudel, A., Allaart, M., Ancellet, G., et al.: Impact of the COVID-19 economic downturn on tropospheric ozone trends: an uncertainty weighted data synthesis for quantifying regional anomalies above Western North America and Europe, AGU Advances, 3, e2021AV000542, https://doi.org/10.1029/2021AV000542, 2022.

Dunn, R. J. H., Alfred, F., Gobron, N., Miller, J. B., and Willett, K. M.: Global Climate [in "State of the Climate in 2020"], Bulletin of the American Meteorological Society, 102, S11 – S141, https://doi.org/10.1175/BAMS-D-21-0098.1, 2021.

Steinbrecht, W., Kubistin, D., Plass-Dülmer, C., Davies, J., Tarasick, D. W., et al.: COVID-19 crisis reduces free tropospheric ozone across the Northern Hemisphere, Geophysical Research Letters, 48, e2020GL091 987, https://doi.org/10.1029/2020GL091987, 2021.

Ziemke, J. R., Kramarova, N. A., Frith, S. M., Huang, L.-K., Haffner, D. P., et al.: NASA satellite measurements show global-scale reductions in free tropospheric ozone in 2020 and again in 2021 during COVID-19, Geophysical Research Letters, 49, e2022GL098712, https://doi.org/10.1029/2022GL098712, 2022.

**Comment by Rodrigo Seguel:**

*General comments*
The manuscript addresses the impact of the COVID-19 pandemic on the ozone mixing ratio measured at 41 high-altitude surface monitoring stations, mainly in the United States and Western Europe, during 2020 (downturn) and 2021, as the beginning of the global economic recovery. The paper adopts the guidelines suggested by TOAR-II. In this regard, the authors applied quantile regression to estimate trends (Chang et al., 2023), facilitating future comparisons with ongoing papers to be summited in this special issue. The study also analyzes the variability in the ozone column (3-6 km) from the IASI satellite products. Surface negative anomalies, especially in 2020, are consistent with IASI observations and previous publications (Ziemke et al., 2022). Additionally, the study relates wildfires to positive anomalies observed in the Western US. Therefore, the paper is a valuable contribution to the TOAR-II Community Special Issue.
*We thank Dr. Rodrigo Seguel for his valuable suggestions and encouraging evaluation. In the following, we report our point-to-point replies to each of the raised points. Modifications to the text are performed in the revised version of the manuscript and are marked in different colors.*

*Minor comments*
Table 1: The Denali National Park (DEN) station is not exactly a high-elevation site (663 m a.s.l.). However, there is no explicit definition to classify high-elevation sites provided, at least by TOAR-II. Can the authors indicate or clarify the motivation to include this station?
*Despite not being located above 1000 m a.s.l., like most of the other stations presented in this study, the Denali National Park (DEN) site was considered for the length of its time series, and for increasing $O_3$ sampling at high latitudes. Moreover, DEN is an isolated site, still elevated compared to the surrounding regions, and representative of the well mixed boundary layer conditions of a broad region around it, as we consider MDA8 data for this site (see Sect. 2.1.1).*

Line 33: "non-methane volatile organic carbons (NM-VOCs)". Please substitute "carbons" by "compounds," which is the standard definition. Alternatively, one finds in the literature non-methane hydrocarbons (NMHC), which is not accurate because do not involve other heteroatoms present in the chemical structures, such as oxygen and nitrogen, among others.
*Done.*

Line 104, 108: I assume that most of the 41 stations meet the 75% threshold. Is it possible, for instance, to indicate those stations with data availability lower than 75% in Table 1? To what extent are the MKN, MBO, or PDI critical to the analysis? In this regard, I suggest rewording the sentence: "which might suffer from issues that prevented complete data sampling in each month."

*In lines 105-108 we are referring to the thresholds on hourly data availability for obtaining each monthly mean, and we believe that reporting such percentages in Table 1 could be misleading, because they vary along each time series, and reporting, e.g., an average value could not be representative of the whole data availability for each dataset. The term "critical" for MKN, MBO and PDI was referred to data availability only; it is now removed to avoid confusion, and the sentence was reworded.*

Line 208: Can the authors check the reference WMO (2021)? WMO 2021 describes the intense wildfire season of 2020, not "Western North America in 2021".
*Thanks, we were referring to the "WMO Air Quality and Climate Bulletin No. 2 - September 2022". We updated the reference.*

Line 221, 228, 264: The term "bump" is a positive anomaly probably due to transport processes, as stated by the authors. I suggest not using "bump" because it is unclear and can be described using standard terms.
*The term "bump" was substituted by standard terms (e.g., "increase", "positive anomaly").*

**Reviewer #2:**

This manuscript calculates trends and anomalies in ozone concentrations at high elevations stations and explores the response in these metrics due to changes in behavior during the COVID-19 pandemic. This is additionally explored by examining temporal profiles in satellite retrievals of O3 column data.

Overall, the paper is a thorough and well written account of the changes experienced at the chosen sites. Subject to some clarification of the methods used and the specific comments below, this paper should be accepted for publication.

*We thank the reviewer for his/her/their valuable suggestions and encouraging evaluation. In the following, we report our point-to-point replies to each of the raised points. Modifications to the text are performed in the revised version of the manuscript and are marked in different colors.*

*General Comments*

In sections 2.1.2 the authors describe the calculation of the O3 anomalies after detrending and de-seasoning the timeseries. I have two primary concerns with this section:

- While I believe the process described here is sound, my concern is that as written, reproducing the methods from this description is not facile.
- There is substantial mixing of mean averaging with median seeking methods (i.e. quantile regression at 50%).

Restructuring of this section would go a long way to allay these concerns, and I would consider the inclusion of a simple flowchart in the SI making it overtly clear which steps are applied in what order. A good example of the language that is hard to follow surrounds L114: "Last, we used the differences calculated in the previous step…". Are these differences referring to the resulting de-seasonalised timeseries, which makes sense for a trend, or is this referring to the climatological year, from which one could feasibly calculate anomalies – I believe the authors are referring the former, but I hope this illustrates the uncertainty that is introduced throughout this section. Being explicit with the use of "mean" or "median" over "average" would also help the reader keep the steps clear.

*According to the reviewer's suggestion, we partially reworded Sect. 2.1.2, for clarifying the steps followed for calculating the $O_3$ anomalies. We stated again that, in this paper, we use the term "$O_3$ anomalies" for indicating the deseasonalized and detrended monthly means. For this reason, we have opted to use the term "monthly differences" for indicating the month-by-month difference between each monthly mean and its corresponding "climatological month". We also added a simple flowchart (Fig. S1 in the revised Supplementary Material) for elucidating the different steps.*

[Figure]

*Figure S1. Flowchart indicating the steps followed for the calculation of the monthly $O_3$ anomalies, as explained in Sect. 2.1.2.*

On the second point, my major concern is that the climatological year has been calculated via mean averaging, but then timeseries derived using this to remove seasonality have their trends defined by the median (via QR). After Chang et al, 2023 trend analysis using QR is the preferred method here, but I would question why de-seaonalisation was not conducted using a climatological year calculated using the median also, as one would surely want this to be less sensitive to outliers in any given year.

*We thank the reviewer for the feedback. Indeed, ever since Weatherhead et al. (1998) explicitly stated that: "While the seasonal component is essential in practical modeling of geophysical time series, estimation of this component does not have much impact on the statistical properties of the estimates of the other terms in the model (Section 2.1)", most trend studies assume this is the case for atmospheric composition trend detection.*

*A recent study has carried out the exact same analysis tailored to QR (see supplementary Sect. S3 in Chang et al., 2023 for further details). Specifically, they have compared the trends based on different approaches: including (i) data are deseasonalized by the climatological monthly means, then percentile trends (5th, 50th and 95th) are fitted using QR; and (ii) percentile trends and percentile seasonal cycles are jointly fitted using QR (so different seasonal cycles are allowed for different percentiles). They found that the trend estimates are strongly consistent for all percentiles, even if the time series is relatively short (Mt. Bachelor, 2004-2021). More interestingly, even though they found different seasonal*

*peaks for the monthly means and the monthly 5th percentiles at Mt. Bachelor, the resulting 5th trends from both approaches are still consistent (1.9 [±1.4] ppbv/decade for approach (i) and 1.7 [±1.5] ppbv/decade for approach (ii)).*

*However, since only two sites are used in the comparisons (Mt. Bachelor and Mauna Loa) in Chang et al. (2023), and as requested from the reviewer's comment, we revisited the statement from Weatherhead et al. (1998) and carried out additional analysis to demonstrate the sensitivity of trend estimates based on climatological means or medians. The results are shown in Fig. S3: although some differences can be seen at individual sites, the general features and conclusions remain the same.*

*This result is not unexpected and fits the statistical theory. We give a thorough discussion as follows:*

- *In terms of trend analysis, seasonal adjustments have two purposes: (1) if missing values are not evenly distributed over different months, the trends will be improperly weighted and thus prone to bias; and (2) trend uncertainty is very likely to be inflated if the seasonality is not accounted for, because regular seasonal pattern should not be considered to be part of trend uncertainty (Chandler and Scott, 2011). In general, as long as the estimation method serves the above purposes, then it is a valid representation of the seasonal cycle.*
- *Such seasonal adjustments can be considered to be a decomposition of a time series into three components: seasonal, trend and residuals. These three components often assume to be additive and independent (unless complex interaction models are considered) (e.g., Cleveland, 1990; Weatherhead et al., 1998; Carslaw, 2005; Gardner and Dorling, 2000; Boleti et al., 2020; Cooper et al., 2020).*
- *Deseasonalization reduces relative variability between different months, but relative variability within the same month remains unchanged (since the same constant is subtracted, e.g., the January trends are the same before and after deseasonalization). It indicates extreme values will still remain extreme after deseasonalization (albeit the magnitude will change). Generally speaking, we should expect the resulting variability is the same in the mean-based and median based deseasonalized data, because deseasonalization is merely designed for removing regular patterns, the other information conveyed in the original time series (including extreme values) will still be present after the process.*

*Based on the above discussions, although median-based seasonal cycle is less sensitive to extreme values than mean-based seasonal cycle, since the deseasonalized data is expected to convey a similar amount of variability and extreme values, we should expect the resulting trends to be consistent (regardless of least squares methods or QR). On the other hand, since the estimated seasonal cycle might be varying from different approaches, they contribute some differences in trend estimates. In summary, the statement from Weatherhead et al. (1998) does not imply the impact is neglectable, but should be interpreted as no systematic discrepancies should be found between different approaches to estimate the seasonality (as shown in Fig. S3).*

*Therefore, a sentence was added at the end of Sect. 2.1.2: "We also carried out additional analysis to demonstrate the sensitivity of trend estimates based on climatological means or medians. The results are shown in Fig. S3 of the Supplementary Material: although some differences can be seen at individual sites, the general features and conclusions remain the same, indicating that no systematic discrepancies are found between different approaches to estimate the seasonality."*

*Boleti, E., Hueglin, C., Grange, S. K., Prévôt, A. S., & Takahama, S.: Temporal and spatial analysis of ozone concentrations in Europe based on timescale decomposition and a multi-clustering approach, Atmospheric Chemistry and Physics, 20(14), 9051-9066, https://doi.org/10.5194/acp-20-9051-2020, 2020.*

*Carslaw, D. C.: On the changing seasonal cycles and trends of ozone at Mace Head, Ireland. Atmospheric Chemistry and Physics, 5(12), 3441-3450, https://doi.org/10.5194/acp-5-3441-2005, 2005.*

*Chandler, R and Scott, M.: Statistical methods for trend detection and analysis in the environmental sciences. London, UK: John Wiley & Sons, 2011.*

*Chang, K.‑L., Cooper, O. R., Rodriguez, G., Iraci, L. T, Yates, E. L., et al.: Diverging ozone trends above western North America: Boundary layer decreases vs. free tropospheric increases, Journal of Geophysical Research: Atmospheres, 128, e2022JD038090, https://doi.org/10.1029/2022JD038090, 2023.*

*Cleveland, R. B., Cleveland, W. S., McRae, J. E., & Terpenning, I.: STL: A seasonal-trend decomposition procedure based on Loess, Journal of Official Statistics, 6(1), 3-73, 1990.*

*Cooper, O. R., Schultz, M. G., Schröder, S., Chang, K. L., Gaudel, A., et al.: Multi-decadal surface ozone trends at globally distributed remote locations, Elementa: Science of the Anthropocene, 8, 23, https://doi.org/10.1525/elementa.420, 2020.*

*Gardner, M. W., and Dorling, S. R.: Meteorologically adjusted trends in UK daily maximum surface ozone concentrations, Atmospheric Environment, 34(2), 171-176, https://doi.org/10.1016/S1352-2310(99)00315-5, 2000.*

*Weatherhead, E. C., Reinsel, G. C., Tiao, G. C., Meng, X.-L., Choi, D., et al.: Factors affecting the detection of trends: Statistical considerations and applications to environmental data, Journal of Geophysical Research: Atmospheres, 103, D14, 17149-17161, https://doi.org/10.1029/98JD00995, 1998.*

[Figure]

*Figure S3. Same as Fig. 2, but in this case the "climatological year" for obtaining the O₃ anomalies was derived by median averaging.*

*Specific comments*
Line 101 – "The deseasonalization allows to produce a more…" Should perhaps read: "The deseasonalization allows the production of a more…"
*Done.*

Figure 2. – The use of two colours per p-value is not well described. Is hue used to denote trend sign, and saturation for significance? In the previous figure a very similar colour pallet was used to show region, which has now been moved the shape in this figure. If this is the case, the use of a different colour pallet here for significance (one colour only) and allowing the x-axis to denote trend direction would be much clearer.
*In the spirit of collaboration and to allow the TOAR-II findings to be comparable across publications, all manuscripts submitted to the TOAR-II Community Special Issue must meet the guidelines regarding style, units, plotting scales, regional and tropospheric column comparisons, tropopause definitions and best statistical practices. The guidelines are illustrated in this document (https://igacproject.org/sites/default/files/2023-04/TOAR-II_Community_Special_Issue_Gu idelines_202304.pdf). We therefore followed the 7-colors palette (reported in Appendix II) for plotting trend vectors by sign and p-value, in the spirit of collaboration within the TOAR-II Special Issue. For this reason, we could not change the color palette for showing*

*the regions, which are instead identified by the different shapes. The trend direction is already identified by the x-axis, and we also put a vertical line to divide positive and negative trends.*

Figure 3. – Referring the reader to fig. 1 for the definitions of the colours is not good, as the figure + caption should stand alone much more readily. The regions could be added to the y-axis to make the groupings clear.
*We revised Fig. 3 and we now added the region labels to the right of the station names. We changed the caption accordingly. Moreover, the stations in Table 1 and Table S1 were now sorted by region first and then latitude, similar to Fig. 3.*

Line 194, relating to figure 3. The wording here could be changed as I don't think "clearly shows widespread… …in 2020" is strictly true. A similar statement could be made about 2009 or 2015.
*Done.*

Line 217, 2021 falling in the top 5 of 18 years – essentially means 2021 falls in ~ top 1/3rd of years over that period? This could be more clearly phrased.
*The sentence has been rephrased as "ranking in the top one third of years 2003–2021".*

Table 3. This table is attempting to show too much information at once, could be separated out into two to avoid the use of the parenthesis notation, which interrupts being able to read down the columns clearly.
*According to the reviewer's suggestion, we separated Table 3 into two smaller ones, i.e., the revised Table 3 showing the contribution of each sector to the total change ("All sectors"), and Table 4 showing the change of each sector in the selected year with respect to the comparison year.*

**Table 3.** $CO_2$ global emissions variations (expressed in %) from Carbon Monitor (Liu et al., 2020), for the different combinations of years 2019, 2020, and 2021, and with focus on MAM and JJA for each comparison. The percentage represents the contribution of each sector to the total change (i.e., "All sectors").

| Sector | 2020 vs 2019 | | | 2021 vs 2019 | | | 2021 vs 2020 | | |
|---|---|---|---|---|---|---|---|---|---|
| | All | MAM | JJA | All | MAM | JJA | All | MAM | JJA |
| All sectors | −5.3% | −13.6% | −4.0% | +0.5% | +0.9% | +1.5% | +6.1% | +16.8% | +5.7% |
| Power | −1.1% | −3.3% | +0.0% | +1.5% | +1.6% | +2.7% | +2.7% | +5.7% | +2.9% |
| Industry | −0.7% | −3.3% | −0.8% | +0.7% | +1.3% | +0.4% | +1.5% | +5.4% | +1.2% |
| Ground transport | −2.0% | −5.0% | −1.4% | −0.6% | −0.8% | −0.5% | +1.5% | +4.8% | +0.9% |
| Residential | −0.2% | −0.3% | +0.1% | −0.1% | −0.1% | +0.0% | +0.0% | +0.2% | +0.0% |
| Domestic aviation | −0.3% | −0.5% | −0.4% | −0.1% | −0.1% | −0.1% | +0.2% | +0.4% | +0.3% |
| International aviation | −1.0% | −1.2% | −1.5% | −0.9% | −1.0% | −1.0% | +0.1% | +0.2% | +0.5% |

**Table 4.** Same as Table 3, but in this case the percentage indicates the sector change in the selected year with respect to the comparison year.

| Sector | 2020 vs 2019 | | | 2021 vs 2019 | | | 2021 vs 2020 | | |
|---|---|---|---|---|---|---|---|---|---|
| | All | MAM | JJA | All | MAM | JJA | All | MAM | JJA |
| Power | −2.8% | −9.0% | −0.1% | 3.9% | 4.3% | 6.7% | 6.9% | 14.7% | 6.8% |
| Industry | −2.5% | −10.6% | −2.4% | 2.2% | 4.2% | 1.1% | 4.9% | 16.5% | 3.6% |
| Ground transport | −10.9% | −26.1% | −7.3% | −3.1% | −4.2% | −2.7% | 8.8% | 29.6% | 4.9% |
| Residential | −1.6% | −2.9% | 1.0% | −1.2% | −0.9% | 0.2% | 0.4% | 2.1% | −0.8% |
| Domestic aviation | −30.8% | −49.6% | −38.0% | −13.1% | −11.9% | −10.3% | 25.5% | 74.7% | 44.5% |
| International aviation | −56.0% | −67.0% | −71.2% | −48.2% | −58.5% | −48.0% | 17.7% | 25.5% | 80.2% |